# Genetic insights into the connection between pulmonary TB and non-communicable diseases: An integrated analysis of shared genes and potential treatment targets

Amira Mahjabeen[1], Md. Zahid Hasan ®[1], Md. Tanvir Rahman ®[2]*, Md. Aminul Islam[1], Risala Tasin Khan[3], M. Shamim Kaiser[3]

**1** Health Informatics Research Lab, Department of Computer Science and Engineering, Daffodil International University, Dhaka, Bangladesh, **2** Department of Information and Communication Technology, Mawlana Bhashani Science and Technology University, Tangail, Bangladesh, **3** Institute of Information Technology, Jahangirnagar University, Dhaka, Bangladesh

* tanvirrahman@mbstu.ac.bd

**Data Availability Statement:** https://pubmed.ncbi. nlm.nih.gov/30395293/.

## Abstract

### Background

Pulmonary Tuberculosis (PTB) is a significant global health issue due to its high incidence, drug resistance, contagious nature, and impact on people with compromised immune systems. As mentioned by the World Health Organization (WHO), TB is responsible for more global fatalities than any other infectious illness. On the other side, WHO also claims that noncommunicable diseases (NCDs) kill 41 million people yearly worldwide. In this regard, several studies suggest that PTB and NCDs are linked in various ways and that people with PTB are more likely to acquire NCDs. At the same time, NCDs can increase susceptibility to active TB infection. Furthermore, because of potential drug interactions and therapeutic challenges, treating individuals with both PTB and NCDs can be difficult. This study focuses on seven NCDs (lung cancer (LC), diabetes mellitus (DM), Parkinson's disease (PD), silicosis (SI), chronic kidney disease (CKD), cardiovascular disease (CVD), and rheumatoid arthritis (RA)) and rigorously presents the genetic relationship with PTB regarding shared genes and outlines possible treatment plans.

### Objectives

BlueThis study aims to identify the drug components that can regulate abnormal gene expression in NCDs. The study will reveal hub genes, potential biomarkers, and drug components associated with hub genes through statistical measures. This will contribute to targeted therapeutic interventions.

### Methods

Numerous investigations, including protein-protein interaction (PPI), gene regulatory network (GRN), enrichment analysis, physical interaction, and protein-chemical interaction,

**Funding:** The author(s) received no specific funding for this work.

have been carried out to demonstrate the genetic correlation between PTB and NCDs. During the study, nine shared genes such as TNF, IL10, NLRP3, IL18, IFNG, HMGB1, CXCL8, IL17A, and NFKB1 were discovered between TB and the above-mentioned NCDs, and five hub genes (NFKB1, TNF, CXCL8, NLRP3, and IL10) were selected based on degree values.

## Results and conclusion

In this study, we found that all of the hub genes are linked with the 10 drug components, and it was observed that aspirin CTD 00005447 was mostly associated with all the other hub genes. This bio-informatics study may help researchers better understand the cause of PTB and its relationship with NCDs, and eventually, this can lead to exploring effective treatment plans.

## Introduction

PTB is a contagious infection caused by the bacillus Mycobacterium Tuberculosis. More than a million people die each year from TB, where smoking, alcohol use, and diabetes all enhance the risk of TB by developing immunodeficiency [1]. Now, it has become the 13th major cause of mortality worldwide and the second largest infectious killer [2]. According to the WHO, 10 million individuals got TB in 2017, and 1.6 million died from it, including 0.3 million HIV-positive persons. Even today, TB is responsible for more deaths than any other infectious disease [3]. These global figures also impacted Bangladesh, where 3.6% of its population, which is roughly 10.0 million people, were afflicted with TB in 2019 [4]. Furthermore, one-quarter of all TB cases globally were also caused by malnutrition [5]. On the other side, TB may weaken the immune system, leaving a person more vulnerable to non-communicable diseases (e.g., diabetes, cardiovascular diseases, cancer, chronic respiratory diseases, and so on), which have the highest global death and morbidity rates. In addition, patients with TB are also prone to have mental health issues [6, 7].

Noncommunicable diseases (NCDs), frequently characterized as chronic illnesses, persist over extended periods due to various contributing factors, including genetic, physiological, environmental, and behavioural influences. The primary categories of NCDs encompass cardiovascular diseases (e.g., heart attacks and strokes), cancers, chronic respiratory conditions (notably asthma and chronic obstructive pulmonary disease), and diabetes. A notable aspect of the global impact of NCDs is their disproportionate prevalence in low- and middle-income nations, where they account for over three-quarters of NCD-related deaths globally, translating to approximately 31.4 million fatalities. This statistic underscores the need for focused health strategies in these regions to address the growing NCD burden [8]. Furthermore, NCDs are lifestyle illnesses that have become more prevalent as a result of globalization and economic expansion. According to WHO, lifestyle diseases are the primary cause of death and disability, as well as the most significant barrier to global development [9–13]. The government and other groups are working together to provide low-cost solutions to minimize common modifiable risk factors where lifestyle adjustments may help to avoid these diseases [14].

Among the NCDs, Cancer is one of the top reasons for mortality across the world, accounting for 15 million new cancer diagnoses and 8.2 million deaths per year. Lung cancer (LC) ranks as the most prevalent malignancy in both men and women, and at least 1.8 million

individuals die from lung cancer every year [15, 16]. However, the Fragile Histidine Triad (FHIT) gene mutation is also linked to lung cancer in patients with PTB [17]. Silicosis (SL) is another risk factor for TB. It is a kind of pneumonitis that affects the lungs due to large amounts of silica dust being breathed in. Researchers suggest that PTB is more likely to affect SL patients than individuals who do not have the condition. However, silica has been linked to Rheumatoid Arthritis (RA) and Chronic Kidney Disease (CKD) [18, 19]. Chronic kidney disease develops when a disease or condition compromises kidney function, causing kidney damage to worsen over time. CKD was the 11th leading cause of mortality globally in 2016, estimated at 11–13%, and became a significant health problem. Patients with CKD lose their renal function, and then the symptoms appear [20]. According to the xNational Institute for Excellence in Health and Care, people with CKD at any stage have a 10 to 25% chance of having active TB. CKD affects up to 16% of the global population, and at least 2% of them require dialysis, which also increases the risk of TB compared to patients who do not require it [21, 22]. Alongside CKD, Rheumatoid arthritis (RA) is a chronic autoimmune condition where the immune system mistakenly attacks healthy cells, leading to painful inflammation in affected areas. Primarily affects the joints, generally attacking many joints at once, and joints are often destroyed, and ligaments and tendons get weaker [23]. RA becomes more common with age, and women have a more considerable risk than males. RA causes joint discomfort, swelling, stiffness, cartilage, and bone degeneration that can impair joint function. Beyond the joints, other organs such as blood vessels, kidneys, heart, lungs, and liver may be affected [24]. RA patients have an elevated chance of contracting life-threatening infections, cancer, and cardiovascular disease [25, 26]. Apart from these, Parkinson's disease (PD) is a neurodegenerative condition characterized by spontaneous or uncontrolled movements, including tremors, rigidity, and issues with balance and coordination. These symptoms generally develop slowly and worsen as the disease progresses. As PD advances, individuals often face increasing challenges in walking and communication, indicative of the disease's progressing impact on both motor and non-motor abilities [27]. The symptoms of PD develop over time but may not be noticeable until the disease has progressed significantly [28]. However, patients with TB had a 1.38-fold increased risk of PD. In the TB cohort, the likelihood of Parkinson's disease declined over time. When treating patients with TB, physicians should be mindful of the possibility of Parkinson's disease [29]. Cardiovascular diseases (CVDs) represent a significant global health challenge, being responsible for an estimated 17.9 million deaths annually. This category encompasses a range of disorders affecting the heart and blood vessels, including coronary heart disease, cerebrovascular disease, and rheumatic heart disease, among others [30].TB may indicate immune dysregulation, predisposing individuals to CVD. The potential link between TB and CVD lies in TB's sustained immune activation characteristic. Tuberculosis-related immune activation and cross-reactivity of antibodies targeting mycobacterial HSP65 with self-antigens in blood vessels may contribute to CVD [31]. Comorbidities like obesity, abnormal lipid profiles, and insulin resistance are often linked to CVD, and hyperlipidemia, diabetes, hypertension, and smoking are linked to CKD [32, 33]. Besides, Diabetes mellitus (DM), a global epidemic that reveals itself when glucose levels in the blood rise too high and kidney function declines, is caused by a failure to create or properly utilize insulin. It was calculated that the disease directly caused 1.5 million deaths in 2019. This disease's incidence is rising rapidly in low and middle-income economies [34]. TB and NCDs can not only coexist but also amplify the risk of one another. TB can compromise immune surveillance, heightening vulnerability to NCDs, which collectively account for two-thirds of global mortality [35]. Because Some NCDs are not curable, and early detection can save lives. LC can be cured if treatment starts at the early stages. However, PD, CKD, RA, DM, and SL are not curable diseases. CVD can be cured, but long-term treatment is needed [23, 36–41].

We aimed to examine the genetic relationship between PTB and seven NCDs (lung cancer, diabetes mellitus, Parkinson's disease, silicosis, chronic kidney disease, cardiovascular disease, and rheumatoid arthritis). Here, a total of nine common genes were discovered. The main key findings of this study are:

- To examine the genetic relationship among seven non-communicable diseases such as LC, DM, PD, SL, CKD, CVD, and RA with PTB.

- To search for drugs that could modify the abnormal gene expression in PTB with non-communicable diseases.

- To analyze the detailed genetic data and reveal hub genes that interact with other genes and chemicals that alter gene networks.

- To discover drug components concerning 5 hub genes based on P-value and adjusted P-value.

- Validate hub genes through RNA-sqn data, where p-value, adjusted p-value $< 0.05$ and log2FoldChange $\geq 0.5$

## Related work & motivation

In past years, observational evidence of a link between cancer and PTB has developed [42]. A study identifies miRNA expression patterns in serum in Egyptian people with LC, TB, and pneumonia. miR-197 and miR-182 levels were high in LC and TB patients, and miR-21, miR-197, and miR-155 levels were high in LC, TB, and pneumonia patients [43]. In patients with CKD, the risk of TB increases in CKD stage 3, and CKD stage 5 is higher than in the number of dialysis patients [44]. In older patients, renal impairment is a common side effect of anti-TB medication [45]. A higher cancer risk at ten different sites has been linked to TB. Low-resource countries disproportionately bear the cancer burden linked to TB [46]. Another study extracted 13 common genes (IL6, TLR4, TNF, CRP, CCL2, IL10, IL1B, TGFB1, ADIPOQ, ACE, VEGFA, IL1RN, HIF1A) among TB, cirrhosis, chronic obstructive pulmonary disease, diabetes mellitus, obesity, ischemic heart disease, ischemic stroke from gene regulatory network, genetic PPI, enrichment analysis, co-expression, and physical interaction. Four significant genes (TNF, IL6, IL10, and IL1B) were used to explore drug design and treatment from protein disease interaction and protein chemical interaction network [47]. From a systematic review, people with hematologic, head and neck, or lung cancer in the US had a 9-fold higher chance of getting active TB. However, most investigations failed to quantify how long they followed up with participants and could not estimate annual TB case risk after a cancer diagnosis or risk [48]. A study showed that 2.79% of people with LC had previously had TB. These individuals had a higher risk of dying within three years of being diagnosed with LC than those without a TB history. However, this study used a claimed database and lacked genetic data for pulmonary function, laboratory, and lung cancer, but this research didn't provide information about how long they followed patients after their cancer diagnosis [49]. In a study, 11 common genes- TNF, IL6, ICAM1, TGFB1, BDNF, AGT, ADIPOQ, CRP, PON1, SOD1, and IL8 were found in 4 diseases (diabetes, kidney disease, stroke, and anxiety). Protein-protein interaction and regulatory interaction networks of the 11 common genes were analyzed. Among the 11 genes, seven significant genes (TGFB1, TNF, PON1, CRP, ICAM1, CXCL8, and AGT) were used for the gene co-expression network and physical interaction pathway [50]. In another retrospective study, patients with RA are more likely to get TB, which could result from anti-TNF

treatments, but very few TB patients constituted a serious constraint that contributed significantly to the possibility of bias [51]. A study finds growing evidence that individuals with—NCDs represent a high-risk group for developing active TB [52]. The current quantitative analysis showed that infection might increase the chance of getting PD, and the significance of the relationship differed based on the particular pathogens. Still, the number of qualified studies was small, and there are different kinds of infections [53]. The association between type 2 diabetes and PD has become clear, but the underlying molecular pathways are still unknown [54]. CVD is the alarming rise of DM and its significant consequences [55]. CVD and PD share biological processes, including inflammation, insulin resistance, oxidative stress, and lipid metabolism [56].

Lifestyle, environmental factors, behaviors, and dietary intake can cause non-communicable diseases. In this regard, a study designed a cross-sectional methodology to determine the NCDs risk factor during the treatment of PTB, but the study may have a desirability bias [57]. It might be challenging to determine whether or not the diseases are connected because it is difficult to make a straight cause-and-effect inference. After all, the scenario could have had different results at different times. Some longitudinal studies analyzed limited health centers and included little information about patients. Many studies have explored the prevalence of NCDs with high PTB incidents, but the association remains unclear. Besides, the impact of PTB on NCDs is not mentioned [58, 59]. Epidemiological data is insufficient to identify the relationship between NCDs and PTB. Some studies included a smaller and narrow dataset wherein the essential risk factors associated with PTB, such as smoking, diabetes, and socioeconomic status, were not taken into consideration [60]. However, the characteristics of PTB may vary in different regions. In this study, we uncovered most of the NCDs as the existing studies have failed to adequately report all of them.

## Methodology

This study has four phases: i) Gene collection, ii) Identification of Hub genes, iii) Analysis of the genes, and iv) Identification of drug candidates. The utilized tools and databases are also described in Table 1.

### Phase—I: Gene collection

The NCBI is an integral component of the National Library of Medicine under the National Institutes of Health purview. Its primary mission is the development of comprehensive molecular biology information systems. This database holds 32928347 records of genome information [69]. This study collects the human genes of PTB and the NCDs (LC, DM, PD, SL, CKD, CVD, and RA) to explore the genetic relationship and discover potential drug candidates. Also, RNA-seq datasets were used to validate the hub genes.

### Phase—II: Identification of hub gene

Hub genes play an important role in drug discovery because they are prominent within biological networks. Understanding and targeting hub genes can provide important insights into the molecular mechanisms behind illnesses, allowing for the creation of targeted therapeutics and medications. Hub genes are also essential in network pharmacology because they allow for a complete knowledge of drug effects in the context of complicated biological relationships.

Our study first established a generic PPI network to identify the hub genes among PTB and NCDs. Here, generic and tissue-specific PPI (whole blood tissue) has been used to analyze protein-protein interaction networks. These two analyses were carried out with the help of NetworkAnalyst, where the STRING Interactome database performed genetic PPI with a

**Table 1. Widely used tools and databases in this study.**

| Item | Description |
|---|---|
| *NetworkAnalyst* (https://www.networkanalyst.ca/) | NetworkAnalyst is an analytic platform to analyze and find relevant characteristics, patterns, functions, and relationships in complex gene expression data. It provides visual analytics experience for data analysis [61]. |
| *Cytoscape* (https://cytoscape.org/) | It is a widely used open-source program for visualizing gene and protein interaction networks, among other types of biological networks [62]. |
| *CytoHobba* (http://apps.cytoscape.org/apps/cytohubba) | The cytoHobba tool performs network component ranking based on their network-related characteristics. It encompasses an array of 11 topological analyses and 6 centrality measures, which collectively enable the prioritization of nodes within a network [63]. |
| *Enrichr* (https://maayanlab.cloud/Enrichr/) | It offers multiple methodologies for calculating gene set enrichment, and the results can be viewed in interactive ways. It uses different machine learning methods. It is a search engine for thousands of annotated gene sets [64]. |
| *GeneMANIA* (https://apps.cytoscape.org/apps/GeneMania) | GeneMANIA is used for a wide range of inquiries about genomic, proteomic, and gene function information. It measures gene function hypotheses, list analysis, and gene prioritization for functional experiments [65]. |
| *String* (https://string-db.org/) | The STRING database tracks physical and functional protein-protein interactions. Which may cooperate in a metabolic or signaling route, control each other through intermediates, or contribute to a cellular structure. Data sources include automated text mining, computational interaction predictions, genetic context, interaction experiment databases, and curated complexes/pathways [66]. |
| *Stitch* (http://stitch.embl.de) | It's a repository for protein-chemical interactions that compiles data from various sources, including experiments, human curation, computational predictions, and text mining. It contains 390,000 chemicals and 3.6 million proteins from 1133 species [67]. |
| *ENCODE* (https://www.encodeproject.org/) | The ENCODE (Encyclopedia of DNA Elements) database is a comprehensive resource that provides valuable information on the functional elements of the human genome, including regulatory regions, protein-coding genes, and non-coding RNAs. This database remains indispensable for researchers seeking to unravel the complexities of genome function and its implications in various biological processes and diseases [68]. |

900-confidence score cutoff. Here, STRING confidence scores indicate the likelihood of finding related proteins in the same pathway. The DifferentialNet database, which offers distinct interactomes for over 29 human tissues, was used to examine the tissue-specific PPI. It reveals tissue-specific protein functions, processes, and phenotypes. In tissue-specific PPI, whole blood cells have been counted with filter 15.0. In this regard, the acquired genetic PPIs are analyzed using Cytoscape to provide a straightforward graphical illustration of the network and explore biological interaction.

At this stage, the cytoHobba plugin is used to identify the hub genes based on topological analysis and centralities. In this context, the degree method is utilized to determine the potential hub genes among PTB and NCDs. The degree method helps determine the closest neighbors (proteins) of a gene. The equation of the degree method (Deg(v)) is as follows:

$$\text{Deg(v)} = |N(v)| \tag{1}$$

Here, $v$ indicates a node, and $N(v)$ stands for its neighbor set [62].

## Phase—III: Gene analysis

In this research, several investigations are carried out to find the key relationship between PTB and NCDs. The analysis includes building protein-protein interaction networks, gene regulatory networks (GRN), enrichment analysis, physical interaction, and protein-chemical interaction has been carried out to demonstrate the genetic correlation between PTB and NCDs. During the study, nine shared genes such as TNF, IL10, NLRP3, IL18, IFNG, HMGB1, CXCL8, IL17A, and NFKB1 were discovered between PTB and the NCDs mentioned above, and five hub genes (NFKB1, TNF, CXCL8, NLRP3, and IL10) were selected based on degree values. Results and conclusion: Enrichment techniques reveal a strong correlation between the shared genes and their pivotal roles in the onset and spread of PTB, unveiling their inherent functions.

**Building PPI networks.** PPIs are central in various cellular and organismal processes, building the cellular framework for immunological defense and cellular communication [70]. This study uses generic PPI and tissue-specific PPI (whole blood tissue) to analyze protein-protein interaction networks.

These two analyses were carried out with the help of NetworkAnalyst, where the STRING Interactome database performed genetic PPI with a 900-confidence score cutoff. Here, STRING confidence scores indicate the likelihood of finding related proteins in the same pathway. The DifferentialNet database, which offers distinct interactomes for over 29 human tissues, was used to examine the tissue-specific PPI. It reveals tissue-specific protein functions, processes, and phenotypes. In tissue-specific PPI, whole blood cells have been counted with filter 15.0. In this regard, the acquired genetic PPIs are analyzed using Cytoscape to provide a clear graphical illustration of the network and explore biological interaction.

**Generating Gene regulatory networks (GRN).** GRN offers a mathematical framework for describing the intricate interaction between gene transcription, genes, and gene products [71]. In this study, we analyzed gene-miRNA and TF-gene interactions to generate the GRN.

miRNAs are noncoding RNAs with 18 to 26 nucleotides, and they pair with specific mRNAs to control gene translation and significantly change many healthy and unhealthy processes. In this regard, the miRTarBase database has been used to find the gene-miRNA interaction. The miRTarBase database has over 13,389 papers with experimental data supporting miRNA–target interactions involving 27,172 target genes from 37 species [72]. On the other hand, the ENCODE database has been used to explore TF-gene interaction. These two analyses are also carried out using the NetworkAnalyst tool.

**Analysis of Gene Ontology and Pathway Enrichment.** Gene set enrichment analysis is a statistical and computational method to determine if a group of genes exhibits statistical importance across a range of biological contexts [73]. Gene Ontology (GO) offers organized, controlled vocabularies and categories covering multiple areas of a gene's molecular and cellular biology. It offers three distinct areas of molecular biology—biological process, molecular function, and cellular component to identify the gene features [74].

For pathway analysis, BioCarta, KEGG, WikiPathways, and Reactome databases were employed [75, 76]. Pathway Enrichment is a vital bioinformatics technique utilized to identify the enrichment of specific biological pathways within a set of genes. These pathways provide insights into the molecular functions and roles of diverse genes. It plays a pivotal role in elucidating the functional significance of gene lists, contributing to a deeper understanding of underlying biological processes and mechanisms [77]. The results from the Pathway analysis and GO terms are explored using Enrichr.

**Generating co-expression, physical and chemical interaction networks.** The co-expression and Physical Interaction Network prediction was accomplished by using GeneMANIA

plugin. Coexpression results were acquired using the STRING database based on the protein co-regulation and RNA expressions. The chemical association network is also generated using the Stitch database.

## Phase—IV: Identification of drug candidates

The most important part of the ongoing research is finding the drug molecules among the eight diseases. The Drug Signatures database (DSigDB) holds 22527 gene sets with 17389 distinct compounds encompassing 19531 genes. It includes gene sets that relate the drugs and small molecules with the gene expression data obtained upon drug administration [78]. The Enrichr platform is used to get into the DSigDB database.

## Phase—V: Gene validation

To validate the selected genes, RNA-seq datasets from NCBI were utilized. For each disease, multiple datasets were chosen to identify up-regulated genes and verify our selected genes. Specifically, for Pulmonary Tuberculosis (PTB), the datasets GSE54992 [79] and GSE19442 [80], which include healthy and case samples, were analyzed. For Parkinson's Disease (PD), GSE20295 [81] and GSE22491 [82] datasets were selected. In the case of Rheumatoid Arthritis (RA), the GSE23561 [83] and GSE157047 [84] datasets were used. For Chronic Kidney Disease (CKD), the datasets GSE66494 [85], GSE15072 [86], and GSE141295 [87] were examined. Cerebrovascular Vascular Disease (CVD) analysis involved datasets GSE51878 [88] and GSE141910 [89]. For Lung Cancer (LC), datasets GSE42826 [90] and GSE30219 [91] were analyzed. Diabetes Mellitus (DM) validation utilized datasets GSE92724 [92] and GSE236746 [93]. The characteristics of these datasets, including the presence of healthy and case samples, are detailed in Table 2.

## Results and analysis

In our research, we discovered 8825 genes for PTB and NCDs (LC, DM, PD, SL, CKD, CVD, and RA). After filtering the genes, we found nine common genes such as: TNF (Entrez ID: 7124), IL10 (Entrez ID: 3586), NLRP3 (Entrez ID: 114548), IL18 (Entrez ID: 3606), IFNG

**Table 2. Dataset attributes used in this study.**

| Disease Name | Accession No | Source | Platform | Sample(Healthy:Case) |
|---|---|---|---|---|
| PTB | GSE54992 | PBMC | GPL570 | 6:19 |
| | GSE19442 | Whole Blood | GPL6947 | 19:32 |
| PD | GSE20295 | Postmortem brain | GPL96 | 53:40 |
| | GSE22491 | peripheral blood | GPL6480 | 8:10 |
| RA | GSE157047 | Blood, Sinovial Fluid | GPL16791 | 12:21 |
| | GSE23561 | peripheral blood, Cell line | GPL10775 | 9:6 |
| CKD | GSE66494 | Kidney | GPL6480 | 8:53 |
| | GSE15072 | PBMC | GPL96 | 8:12 |
| | GSE141295 | Microdissected kidney glomerulus | GPL16791 | 10:14 |
| CVD | GSE51878 | human coronary artery smooth muscle cells | GPL11154 | 6:3 |
| | GSE141910 | Left Ventricle | GPL16791 | 161:161 |
| LC | GSE42826 | Whole Blood | GPL10558 | 52:8 |
| | GSE30219 | Lung Tumor, Non Tumoral Lung | GPL570 | 14:293 |
| DM | GSE92724 | dermal blood endothelial cell | GPL20301 | 6:4 |
| | GSE236746 | Torn rotator cuff tendon | GPL24676 | 3:3 |

(Entrez ID: 3458), HMGB1 (Entrez ID: 3146), CXCL8 (Entrez ID: 3576), IL17A (Entrez ID: 3605), and NFKB1 (Entrez ID: 4790) while analyzing PPIs network, hub genes, GO and pathway analysis, and GRN. Detailed information on these nine genes is presented in Table 3, while for humans, there are 4052 genes for LC, 201 genes for PTB, 1475 genes for RA, 552 genes for CVD, 605 genes for CKD, 58 genes for SI, 789 genes for PD, and 1093 genes for DM [62] as shown in Fig 1. Fig 2 presents a Venn diagram of genes associated with PTB and NCDs to showcase the overlapping gene.

At this moment, we move to identify the hub genes from the Generic PPI with the STRING Interactome database using protein-protein interactions. Here, we discover two subnetworks where Subnetwork1 (the first subnetwork) consists of 214 proteins (marked in blue) and seven genes (NFKB1, TNF, HMGB1, NLRP3, CXCL8, IFNG, IL10), which are marked in orange as shown in Fig 3(a). On the other hand, the subnetwork2 (second subnetwork) has one gene (IL18), marked in orange, and five proteins marked in blue, as also depicted in Fig 3(b). However, we find RELA to be a highly connected protein with CXCL8, HMGB1, NFKBI, and TNF genes.

**Table 3. Description of the discovered common genes as per NCBI.**

| Serial | Gene Symbol | Title of the Gene | Gene Description |
|---|---|---|---|
| 1 | HMGB1 | high mobility group box 1 | This gene produces a High Mobility Group-box protein. This protein affects inflammation, cell differentiation, and tumor cell migration. Several gene pseudogenes have been found. Multiple transcript variants encode the same protein due to alternative splicing. |
| 2 | IL10 | interleukin 10 | This gene encodes a monocyte- and lymphocyte-produced cytokine to influence inflammation and immunoregulation. It boosts B cell survival, proliferation, and antibody production. This cytokine regulates JAK-STAT signaling and blocks NF-$\kappa$B. Mutations in this gene enhance HIV-1 and RA risk. |
| 3 | IL17A | interleukin 17A | It encodes a proinflammatory cytokine generated by activated T cells and is one of five members of the IL-17 receptor family. The cytokine regulates NF-$\kappa$B and mitogen-activated protein kinases, increasing IL6 and PTGS2/COX-2 expression. Cancer, viral illnesses, and inflammatory and autoimmune disorders depend on IL-17A. Rheumatoid arthritis, psoriasis, and multiple sclerosis are linked to high cytokine levels and increased inflammation. |
| 4 | IL18 | interleukin 18 | This gene produces a proinflammatory IL-1 family cytokine constitutively present as a precursor in macrophages and keratinocytes. Caspase-1 converted the dormant IL-18 precursor into an active version that stimulates interferon gamma production and regulates Th1 and Th2 responses. A cytokine storm can be lethal, as this cytokine can damage organs. |
| 5 | NLRP3 | NLR family pyrin domain containing 3 | This protein connects with the NLRP3 inflammasome complex member PYCARD/ASC, which recruits caspases. This complex regulates inflammation, immunological response, and apoptosis by activating NF-$\kappa$B signaling upstream. Through macrophage ion channel creation, the transmembrane pore-forming viroporin SARS-CoV 3a activates the NLRP3 inflammasome. |
| 6 | TNF | tumor necrosis factor | This gene produces a multifunctional proinflammatory TNF superfamily cytokine. This cytokine regulates cell proliferation, differentiation, death, lipid metabolism, and coagulation. It is linked to autoimmune illnesses, insulin resistance, psoriasis, RA, ankylosing spondylitis, TB, autosomal dominant polycystic kidney disease, and cancer. This gene mutation affects cerebral malaria, septic shock, and Alzheimer's disease risk. |
| 7 | IFNG | interferon gamma | This gene produces a type II interferon-like soluble cytokine. The encoded protein is released by innate and adaptive immune cells. The homodimer active protein interacts with the interferon-gamma receptor to activate cells against viral and microbial infections. This gene mutation increases vulnerability to viral, bacterial, parasitic, and autoimmune disorders. |
| 8 | CXCL8 | C-X-C motif chemokine ligand 8 | The CXC chemokine family protein produced by this gene mediates the inflammatory response. This chemotactic factor guides neutrophils to the infection site. IL-8 is part of the proinflammatory signaling cascade with other cytokines and contributes to SIRS. This gene may contribute to the genetics of bronchiolitis, a common respiratory syncytial virus-caused respiratory illness. One gene cluster on chromosome 4q contains this gene and additional CXC chemokine genes |
| 9 | NFKB1 | nuclear factor kappa B subunit 1 | Cytokines, oxidant-free radicals, UV irradiation, and bacterial or viral products activate in NF$\kappa$B, a transcription regulator. NF$\kappa$B translocates into the nucleus and activates several biological activity genes. Inflammatory disorders are linked to NF$\kappa$B activation, whereas prolonged inhibition causes immune cell failure or delayed cell growth. Besides, the response to rapid viral infection is regulated by NF$\kappa$B. |

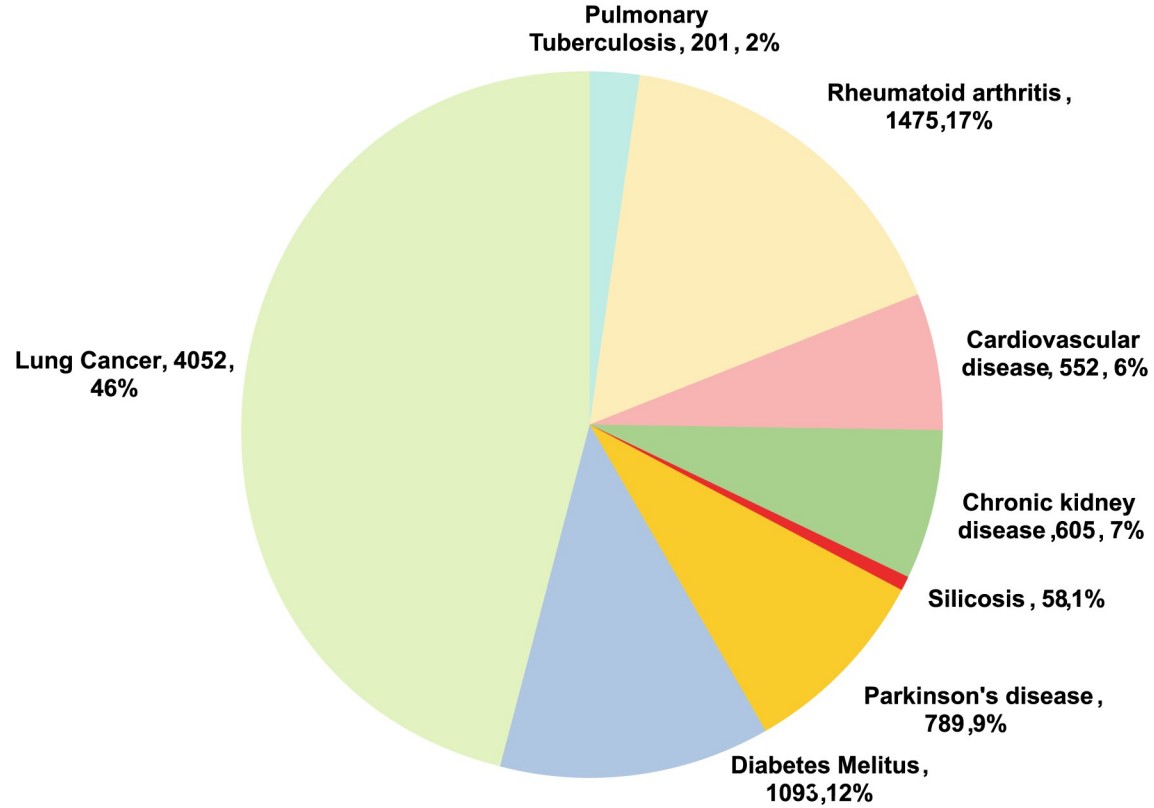

**Fig 1. The pie chart shows the genes of eight diseases in different segments.** The numerical value beside the name of the disease represents the number of genes and their corresponding percentage.

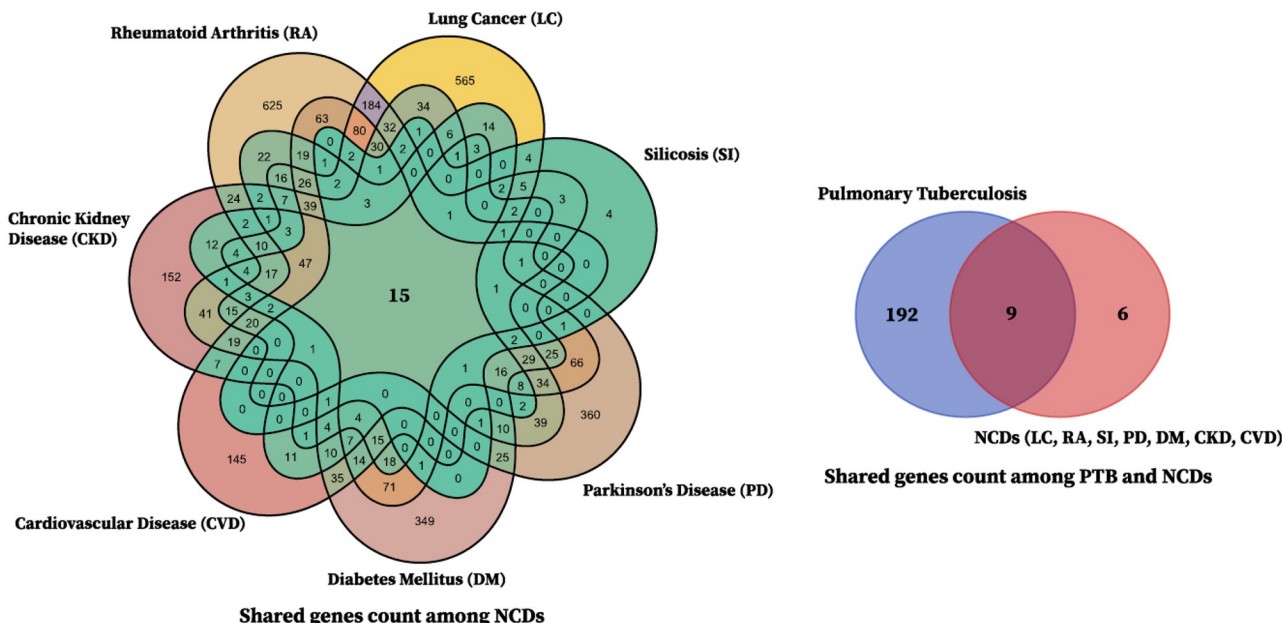

**Fig 2. Common genes among the diseases.** Venn diagram shows the shared genes across PTB and NCDs.

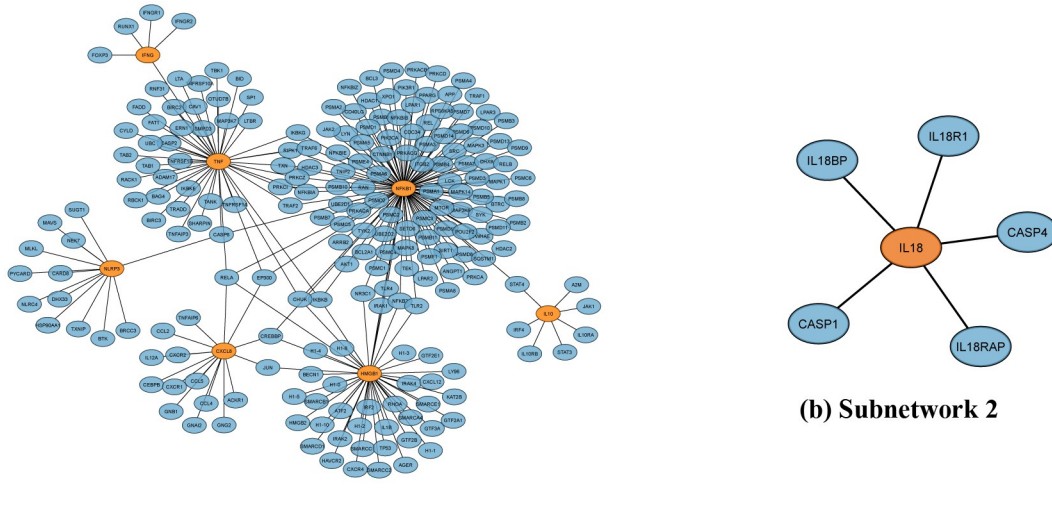

**(a) Subnetwork 1**

**(b) Subnetwork 2**

**Fig 3. Generic PPIs network for the eight diseases.** Orange seeds represent common genes, and edges indicate the connections between the proteins of (a) subnetwork1 and (b) subnetwork2.

At the moment, we also identified five significant hub genes (TNF, NFKB1, HMGB1, CXCL8, NLRP3) from the generic PPI according to the degree value of the genes where the degree value was greater than 10. Table 4 shows that NFKB1 has the highest degree value of 115, a 47-degree value for TNF, HMGB1 had a degree value of 44, and CXCL8 and NLRP3 had 16 and 13-degree values, respectively. Fig 4 demonstrates the hub protein with other inter-acted proteins from the generic PPIs network. Here, we can identify the highly connected genes from degree value, and this method focuses on gene connectivity in the overall network.

Furthermore, Fig 5 represents the tissue-specific PPI network with four subnetworks. It was analyzed by whole blood tissue. Inside Fig 5, subnetwork1 with five genes (CXCL8, NFKB1, TNF, IFNG, and IL10) and 165 proteins are shown in Fig 5(a). Fig 5(b) shows the subnetwork2 with one gene (HMGB1) and 28 proteins while Fig 5(c) shows subnetwork3 with one gene (NLRP3) and 6 proteins. Finally, the subnetwork3 is also drawn in Fig 5(d), and it also has a single gene (IL18) and three proteins. Here, in all of the sub-figures inside Fig 5, the pink node represents a gene, and the green node represents proteins.

Then, we investigated gene regulatory networks through the analysis of gene-miRNA inter-actions and TF-gene interactions. The Gene-miRNA interaction network, constructed using the miRTarBase database, revealed one subnetwork depicted in Fig 6. This subnetwork fea-tured nine genes (HMGB1, NFKB1, IFNG, IL10, TNF, CXCL8, IL18, IL17A, and NLRP3) interacting with 229 miRNAs. Notably, hsa-mir-34a-5P emerged as a key miRNA

**Table 4. Analyzing topological findings for the best-performing hub genes using Eq (1).**

| Gene | Degree |
|------|--------|
| NFKB1 | 115 |
| TNF | 47 |
| HMGB1 | 44 |
| CXCL8 | 16 |
| NLRP3 | 13 |

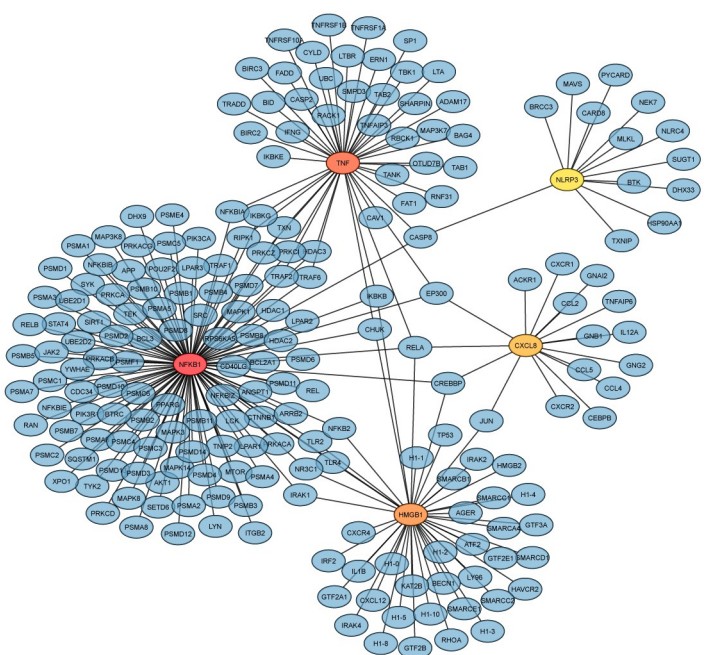

**Fig 4. Hub genes based on degree value.** The highlighted six genes (TNF, NFKB1, HMGB1, CXCL8, NLRP3) are the hub genes among the nine genes from the generic PPI network, connected with protein (blue). The hub genes were identified based on their degree value, where the degree value is greater than 10.

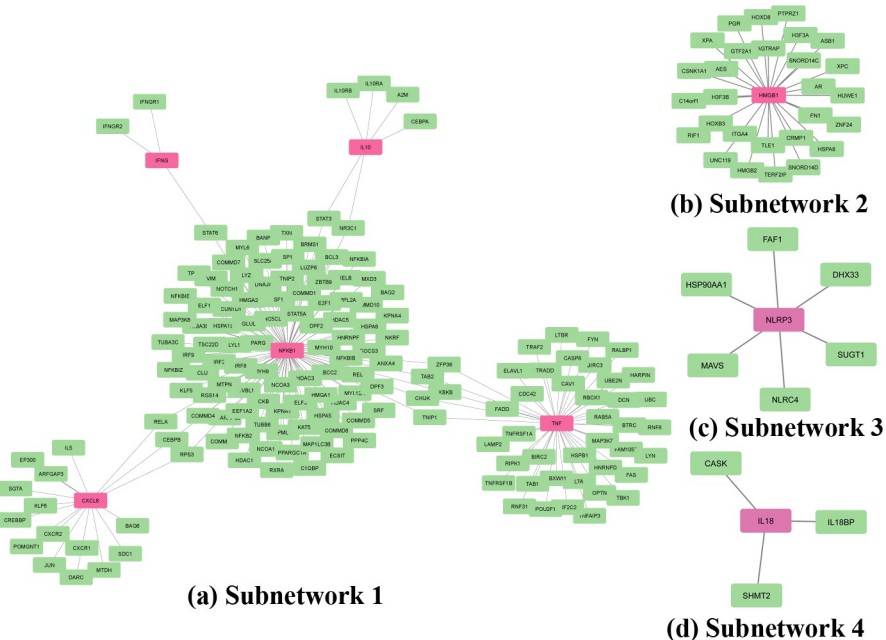

**Fig 5. Tissue-specific PPI network.** The highlighted pink seed denotes the common genes, and the green color indicates the protein that is physically connected with the common genes in (a) subnetowrk1, (b) subnetwork2, (c) subnetwork3, (d) subnetwork4.

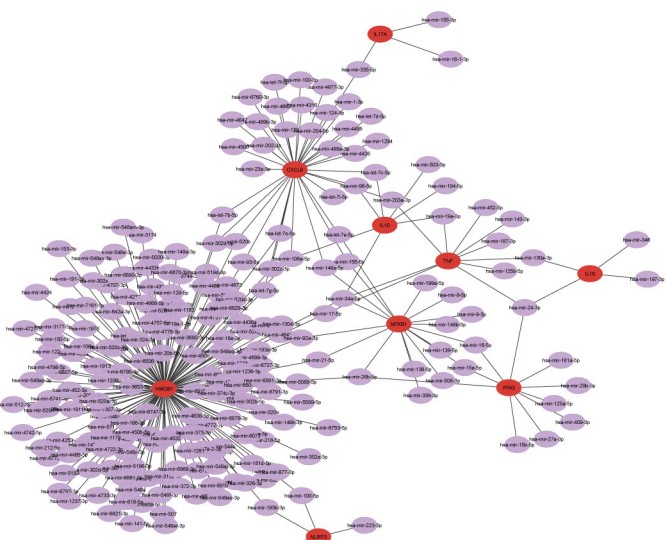

**Fig 6. Gene-miRNA interaction with nine common genes.** The highlighted nine red nodes denote common genes, while the remaining 229 purple nodes are miRNA connected with the genes.

interconnecting TNF, IL10, NFKB1, and HMGB1 genes. Additionally, the TF-gene interaction network analysis unveiled two subnetworks in Fig 7, with subnetwork 1 comprising 54 transcription factors (TFs) linked to six genes (NFKB1, IL10, NLRP3, IFNG, TNF, HMGB1), and subnetwork 2 involving 4 TFs connected to the IL17A gene. These findings provide insights into the intricate regulatory mechanisms governing gene expression.

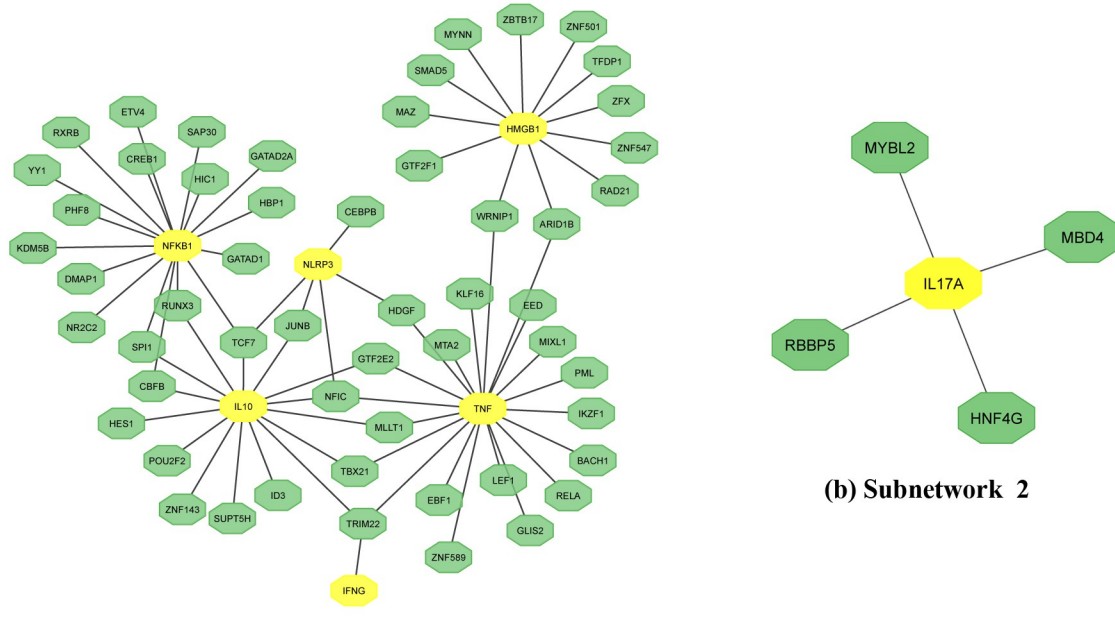

**(a) Subnetwork 1**

**(b) Subnetwork 2**

**Fig 7. TF-gene interaction with common genes.** The highlighted yellow node denotes common genes, while the remaining green nodes are TF, which is connected with the genes.

In the course of our investigation into common genes, an enrichment analysis of the identified shared genes was initiated employing Enrichr. Table 5 delineates data from BioCarta, KEGG, WikiPathways, and Reactome pathways, organized according to their respective P-values. Examination of the data from Table 5 indicates that Yersinia infection, Influenza A, IL-17 signaling pathway, Cytokine-cytokine receptor interaction, and Inflammatory bowel disease exhibit the highest number of gene interactions, as per the KEGG pathway database. Furthermore, the IL-18 signaling pathway (WP4754) manifests the greatest number of associated genes according to WikiPathwas database. As per the Reactome pathway analysis, the pathways connected with the highest number of genes are as follows:

1. Signalling By Interleukins R-HAS-449147

2. Cytokine Signaling In Immune System (R-HSA-1280215)

3. Immune system R-HAS-168256

Concurrently, Table 6 enumerates the top 10 Gene Ontology (GO) keywords for biological processes, molecular functions, and cellular components associated with the common nine genes, arranged based on their respective P-values. Analysis of Table 6 reveals a significant association between the common genes and the biological process of Positive regulation of cytokine production, which is linked with PTB and other diseases, as delineated in Table 2. Besides the biological process, cytokine activity is also connected with the most common gene.

Moreover, Figs 8 and 9 illustrate GO keywords and pathways, respectively, organized by their respective P-value scores. These visual representations provide a comprehensive overview of the significance and relationships of the identified common genes.

At this stage, in order to identify Co-expression and Physical Interaction Networks among genes, we utilized the Cytoscape plugin, GeneMANIA. Here, co-expression and physical interaction networks were constructed using GeneMANIA as a part of the methodology used in this research. The graphical representation in Fig 10 elucidates the co-expression and physical interaction dynamics among related genes, revealing percentages of 56.57% and 17.84%, respectively. Furthermore, the analysis uncovered additional intricate relationships among the identified genes, including co-localization (6.18%), shared protein domains (1.00%), predicted interactions (4.71%), and pathway associations (13.72%). These findings, obtained through the GeneMANIA platform, contribute essential insights into the complex interplay of genes, emphasizing co-expression patterns and physical interactions within the studied biological system.

From Fig 11, the investigation of RNA expression patterns and protein co-regulation provides critical insights into molecular interactions within Homo sapiens. Notable associations have been identified by utilizing coexpression scores. For instance, the CXCL8 and NFKB1 genes exhibit a coexpression score of 0.118, highlighting their closely coordinated regulation. Similarly, CXCL8 demonstrates coexpression scores of 0.088 and 0.095 with IL18 and NLRP3, respectively, indicating potential functional relationships. The IL10 gene reveals coexpression scores of 0.097 and 0.090 with TNF and NLRP3. Additionally, IL17A and IFNG exhibit a coexpression score of 0.099, suggesting a potential synergistic role in immune responses. Noteworthy associations are also observed between NFKB1 and TNF (coexpression score: 0.108), TNF and IFNG (coexpression score: 0.152), and TNF and NLRP3 (coexpression score: 0.146), indicating intricate regulatory networks. Furthermore, IL18 and NLRP3 display a coexpression score of 0.085, while NLRP3 and IFNG exhibit a coexpression score of 0.058. The interaction between HMGB1 and NFKB1 in protein co-regulation stands out with a protein coregulation score of 0.042.

**Table 5. Key pathways identified: Analysis of top pathways in KEGG, Wikipathways, BioCarta, and Reactome databases, including corresponding P-values and genes.**

| Database | Pathways | P-values | Genes |
|---|---|---|---|
| *KEGG* | Inflammatory bowel disease | 7.75E-14 | IL10, IFNG, IL18, TNF, NFKB1, IL17A |
| | Yersinia infection | 7.64E-12 | IL10, CXCL8, IL18, NLRP3, TNF, NFKB1 |
| | Malaria | 9.94E-12 | IL10, CXCL8, IFNG, IL18, TNF |
| | Influenza A | 3.05E-11 | CXCL8, IFNG, IL18, NLRP3, TNF, NFKB1 |
| | Pertussis | 8.63E-11 | IL10, CXCL8, NLRP3, TNF, NFKB1 |
| | Rheumatoid arthritis | 2.42E-10 | CXCL8, IFNG, IL18, TNF, IL17A |
| | IL-17 signaling pathway | 2.56E-10 | CXCL8, IFNG, TNF, NFKB1, IL17A |
| | Amoebiasis | 3.87E-10 | IL10, CXCL8, IFNG, TNF, NFKB1 |
| | Chagas disease | 3.87E-10 | IL10, CXCL8, IFNG, TNF, NFKB1 |
| | Cytokine-cytokine receptor interaction | 7.92E-10 | IL10, CXCL8, IFNG, IL18, TNF, IL17A |
| *BioCarta* | IL-10 Anti-inflammatory Signaling Pathway Homo sapiens h il10Pathway | 1.40E-05 | IL10, TNF |
| | NF-kB Signaling Pathway Homo sapiens h nfk b-Pathway | 3.76E-05 | TNF, NFKB1 |
| | NFkB activation by Nontypeable Hemophilus influenzae Homo sapiens h nthi Pathway | 7.26E-05 | CXCL8, TNF |
| | IFN gamma signaling pathway Homo sapiens h ifng Pathway | 0.002697 | IFNG |
| | Visceral Fat Deposits and the Metabolic Syndrome Homo sapiens h vobesityPathway | 0.003595 | TNF |
| | SODD/TNFR1 Signaling Pathway Homo sapiens h soddPathway | 0.004043 | TNF |
| | NO2-dependent IL 12 Pathway in NK cells Homo sapiens h no2il12 Pathway | 0.004043 | IFNG |
| | Apoptotic DNA fragmentation and tissue homeostasis Homo sapiens h DNAfragment Pathway | 0.00494 | HMGB1 |
| | The 4–1BB-dependent immune response Homo sapiens h 41bb Pathway | 0.005836 | IFNG |
| | Stress Induction of HSP Regulation Homo sapiens h hsp27Pathway | 0.006284 | TNF |
| *WikiPathwas* | SARS-CoV-2 innate immunity evasion and cell-specific immune response WP5039 | 1.35E-08 | IL10, CXCL8, TNF, NFKB1 |
| | IL1 and megakaryocytes in obesity WP2865 | 2.00E-10 | IFNG, IL18, NLRP3, NFKB1 |
| | IL-18 signaling pathway WP4754 | 4.86E-10 | IL10, CXCL8, IFNG, IL18, TNF, NFKB1 |
| | Development and heterogeneity of the ILC family WP3893 | 6.76E-10 | IFNG, IL18, TNF, IL17A |
| | miRNAs involvement in the immune response in sepsis WP4329 | 1.24E-09 | IL10, CXCL8, TNF, NFKB1 |
| | Novel intracellular components of RIG-I-like receptor (RLR) pathway WP3865 | 9.11E-09 | CXCL8, IFNG, TNF, NFKB1 |
| | Allograft Rejection WP2328 | 1.93E-10 | IL10, CXCL8, IFNG, TNF, IL17A |
| | T-Cell antigen Receptor (TCR) pathway during Staphylococcus aureus infection WP3863 | 1.04E-08 | IL10, IFNG, TNF, NFKB1 |
| | Non-genomic actions of 1,25 dihydroxyvitamin D3 WP4341 | 1.81E-08 | CXCL8, IFNG, TNF, NFKB1 |
| | COVID-19 adverse outcome pathway WP4891 | 2.86E-08 | IL10, CXCL8, TNF |
| *Reactome* | Interleukin-10 Signaling R-HSA-678378 | 2.79E-09 | IL10, CXCL8, IL18, TNF |
| | Signaling By Interleukins R-HSA-449147 | 1.01E-10 | IL10, CXCL8, IFNG, IL18, HMGB1, TNF, NFKB1 |
| | Cytokine Signaling In Immune System R-HSA-1280215 | 2.16E-09 | IL10, CXCL8, IFNG, IL18, HMGB1, TNF, NFKB1 |
| | Immune System R-HSA-168256 | 6.44E-08 | IL10, CXCL8, IFNG, IL18, NLRP3, HMGB1, TNF, NFKB1 |
| | Interleukin-4 And Interleukin-13 Signaling R-HSA-6785807 | 9.56E-08 | IL10, CXCL8, IL18, TNF |
| | Cell Recruitment (Pro-Inflammatory Response) R-HSA-9664424 | 1.44E-07 | IL18, NLRP3, NFKB1 |
| | Leishmania Infection R-HSA-9658195 | 2.72E-06 | IL10, IL18, NLRP3, NFKB1 |
| | Interleukin-1 Processing R-HSA-448706 | 6.47E-06 | IL18, NFKB1 |
| | NLRP3 Inflammasome R-HSA-844456 | 2.15E-05 | NLRP3, NFKB1 |
| | Interleukin-1 Family Signaling R-HSA-446652 | 3.50E-05 | IL18, HMGB1, NFKB1 |

**Table 6. GO pathways with their respective P-values and genes for common genes.**

| Category | GO ID | GO Pathways | P-values | Genes |
|---|---|---|---|---|
| *GO biological process* | GO:0071222 | Cellular response to lipopolysaccharide | 8.35E-15 | IL10, CXCL8, IL18, NLRP3, HMGB1, TNF, NFKB1 |
| | GO:0001819 | Positive regulation of cytokine production | 5.06E-15 | IL10, IFNG, IL18, NLRP3, HMGB1, TNF, NFKB1, IL17A |
| | GO:0006954 | Inflammatory response | 8.57E-13 | CXCL8, IFNG, IL18, NLRP3, HMGB1, TNF, NFKB1 |
| | GO:0032722 | Positive regulation of chemokine production | 1.35E-11 | IFNG, IL18, HMGB1, TNF, IL17A |
| | GO:0032732 | Positive regulation of interleukin-1 production | 3.03E-11 | IFNG, NLRP3, HMGB1, TNF, IL17A |
| | GO:0032675 | Regulation of interleukin-6 production | 5.68E-10 | IL10, IFNG, HMGB1, TNF, IL17A |
| | GO:0071219 | Cellular response to molecule of bacterial origin | 7.12E-10 | IL10, CXCL8, NLRP3, HMGB1, NFKB1 |
| | GO:0019221 | Cytokine-mediated signaling pathway | 9.18E-10 | IL10, CXCL8, IFNG, IL18, TNF, NFKB1, IL17A |
| *GO Molecular Function* | GO:0005125 | cytokine activity | 3.16E-11 | IL10, CXCL8, IFNG, IL18, HMGB1, TNF |
| | GO:0060556 | Regulation of vitamin D biosynthetic process | 1.26E-09 | IFNG, TNF, NFKB1 |
| | GO:0060558 | Regulation of calcidiol 1-monooxygenase activity | 2.20E-09 | IFNG, TNF, NFKB1 |
| | GO:0048018 | receptor ligand activity | 9.88E-08 | IL10, IFNG, IL18, HMGB1, TNF |
| | GO:0001067 | transcription regulatory region nucleic acid binding | 9.41E-05 | HMGB1, TNF, NFKB1 |
| | GO:0005126 | cytokine receptor binding | 9.59E-04 | IL10, IL18 |
| | GO:0000976 | transcription cis-regulatory region binding | 0.00152706 | HMGB1, TNF, NFKB1 |
| | GO:0019958 | C-X-C chemokine binding | 0.002248161 | HMGB1 |
| | GO:0097100 | supercoiled DNA binding | 0.002697256 | HMGB1 |
| | GO:1990837 | sequence-specific double-stranded DNA binding | 0.003213257 | HMGB1, TNF, NFKB1 |
| | GO:0000405 | bubble DNA binding | 0.003594909 | HMGB1 |
| | GO:0140297 | DNA-binding transcription factor binding | 0.003693126 | NLRP3, HMGB1 |
| *GO Cellular Component* | GO:0060205 | cytoplasmic vesicle lumen | 0.001149208 | HMGB1, NFKB1 |
| | GO:0034774 | secretory granule lumen | 0.008325128 | HMGB1, NFKB1 |
| | GO:0000793 | condensed chromosome | 0.024043797 | HMGB1 |
| | GO:0035580 | specific granule lumen | 0.027561797 | NFKB1 |
| | GO:1904813 | ficolin-1-rich granule lumen | 0.054018044 | HMGB1 |
| | GO:0055037 | recycling endosome | 0.063401467 | TNF |
| | GO:0005694 | chromosome | 0.069751729 | HMGB1 |
| | GO:0042581 | specific granule | 0.069751729 | NFKB1 |
| | GO:0045121 | membrane raft | 0.071017179 | TNF |
| | GO:0101002 | ficolin-1-rich granule | 0.079832569 | HMGB1 |

Fig 12 illustrates the chemical interaction network. IFNG shows a robust affinity for MgATP with a significant interaction score of 0.914. Simultaneously, the interaction between Phosphoric acid and IFNG was underscored by an impressive score of 0.986. IFNGR1 establishes connections with MgATP and Phosphoric acid with interaction scores of 0.900 in both cases. IL18 joins the narrative with a notable interaction score of 0.871 when paired with MgATP. In contrast, the interaction between IL10 and Phosphate, while moderate with a score of 0.544, contributes to the intricate chemical landscape. NLRP3 emerges as a focal point, forming a robust association with Silica (0.986) and MgATP (0.988). These interactions underscore the pivotal role of NLRP3 in the network. IL10 takes center stage, strongly associating with Rapamycin (score: 0.985). The interaction between NLRP3 and MgATP is robust, with a score of 0.988. NFKB1 exhibits a significant interaction score of 0.911 with MgATP. Notably, RELA, CHUK, and IKBKB each interaction scores with MgATP 0.918, 0.907, and 0.911, respectively.

In this research, the potential therapeutic compounds are identified from the DSigDB database using Enricher. According to the nine common genes (TNF, HMGB1, NFKB1, CXCL8,

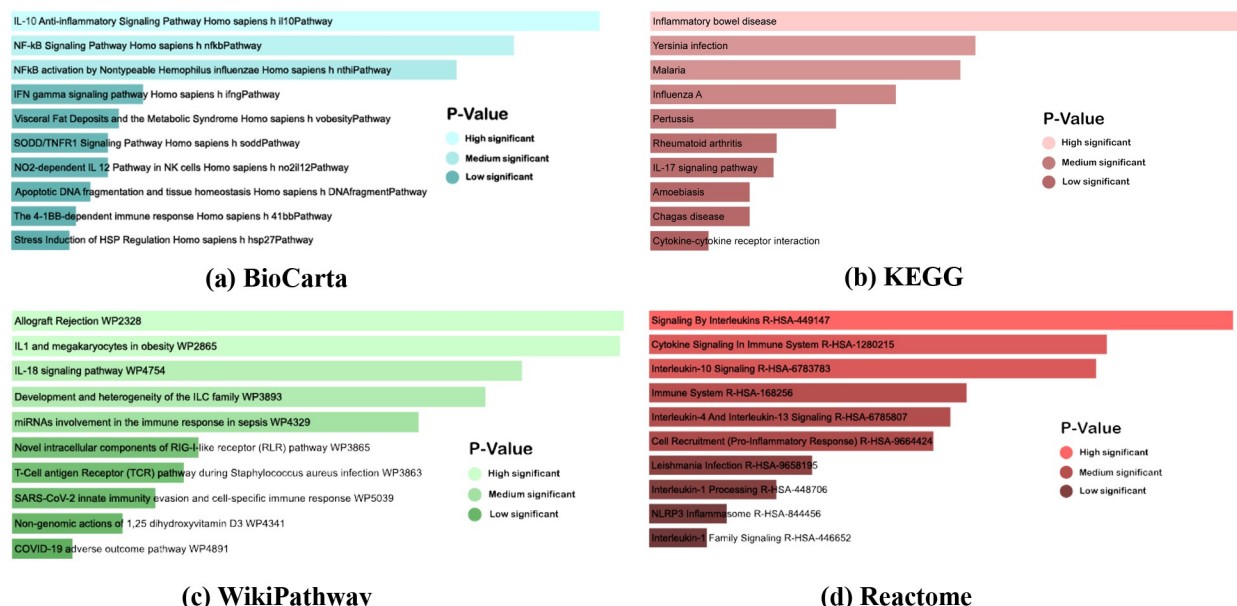

**Fig 8. Pathway analysis of eight diseases.** (a), (b), (c), and (d) depict the pathway analysis from BioCarta, KEGG, Reactome, and WikiPathway according to P-value.

NLRP3, IL10, IFNG, IL18, and IL17A), the drug compounds were suggested based on the P-value and adjusted P-value. Here, we discovered the connection between ten drug compounds and the five hub genes. In this regard, we found bay 11–7082 CTD 00003959, PD 98059 CTD 00003206, and aspirin CTD 00005447 to be the most essential drug compounds as they are

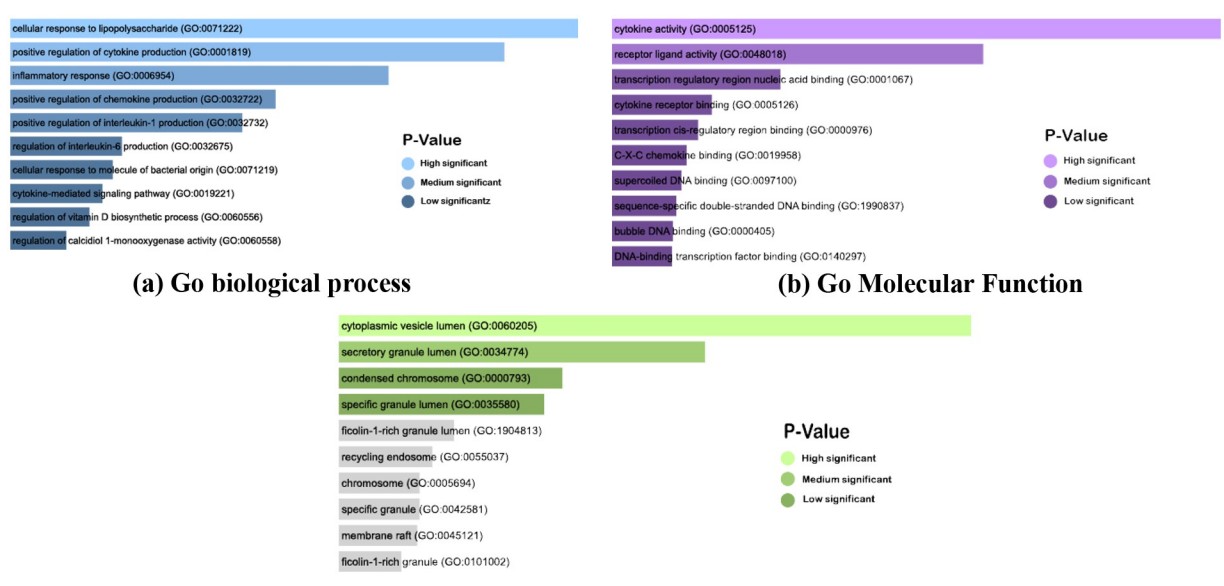

**Fig 9. Gene ontology analysis of eight diseases.** (a), (b), (c), and (d) illustrate the biological process, molecular process, and cellular components, respectively, according to the combined score based on the P-value.

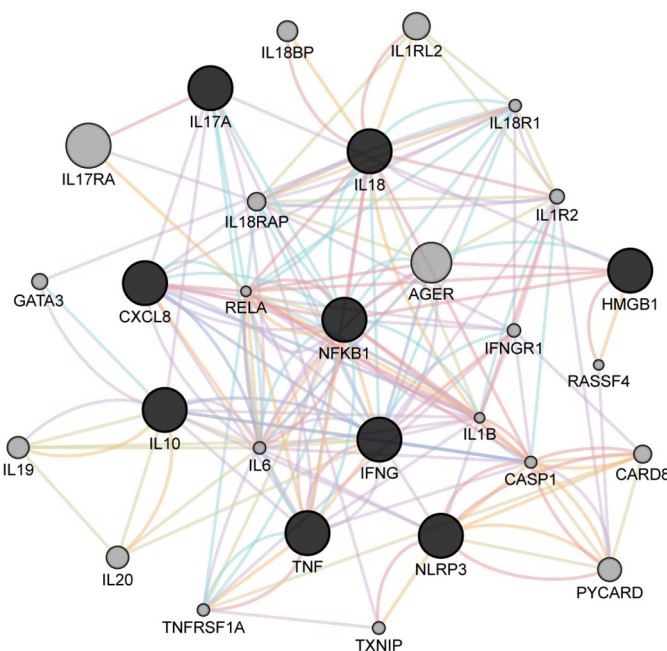

**Fig 10. Co-expression and Physical Interaction Network.** The red line denotes physical interaction. The purple line indicates co-expression, the green line defines genetic interaction, the blue line denotes co-localization, and the orange line denotes predicted.

linked to most of the studied genes. Furthermore, a rigorous analysis of drug targets reveals that the aspirin CTD 00005447 target is mostly related to all the hub genes. However, Table 7 shows all of the predictive drug compounds.

Lastly, RNA sequence data from NCBI was used to validate the hub genes in Tables 8–14. We initially analyze the datasets using the DESeq2 R package. During the analysis, log-fold

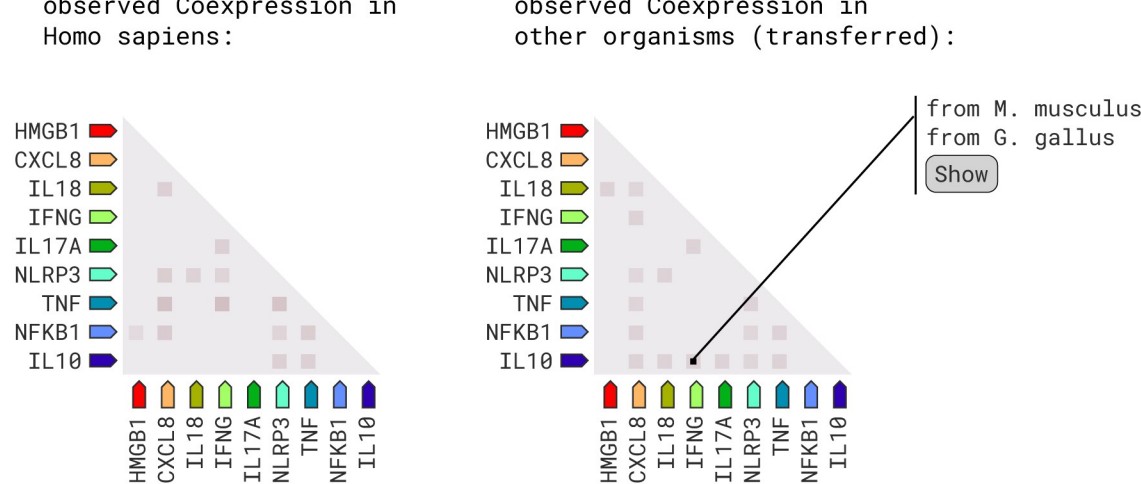

**Fig 11. RNA co-expression scores between proteins.** The heatmap shows the RNA coexpression scores among the nine genes in Homo sapiens and other organisms.

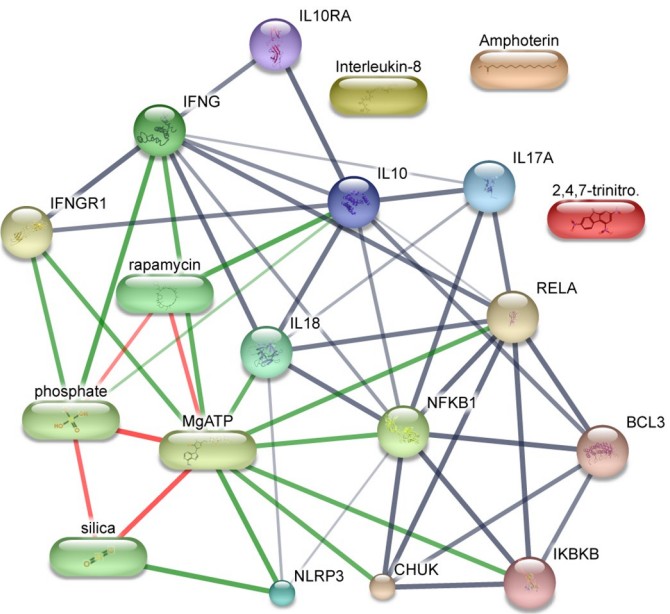

**Fig 12. Chemical interaction network between the nine genes.** The green color edges interconnect between chemicals and proteins. And red color edges interconnect between chemicals.

change (logFC) $\geq$ 0.5 and P-value and adj P-value < 0.05 and a Benjamini–Hochberg method for adjusted p-value.

## Discussion

The major objective of this research was to identify any genetic links that might exist between a variety of disorders. The study analyzes PPIs, GO, pathways, gene regulatory networks, candidate drug detection, physical interaction, and co-expression networks. This analysis shows the correlation between the gene pair. The functional connection among genes in molecular networks has led to a new framework in which it is thought that common and rare diseases are caused by genomic and environmental factors that change whole molecular networks [94].

This study utilized nine common genes to construct generic PPI and tissue-specific PPI networks. PPI networks demonstrate proteomic information about PTB and NCDs. According to

**Table 7. Representing gene count with common symbols in the context of overlapping diseases associated with PTB.**

| Title of the Drug | P-value | Adjusted P-value | Associated Genes |
|---|---|---|---|
| *bay*11 − 7082 *CTD* 00003959 | 5.85E-16 | 9.42E-13 | IL10, CXCL8, IFNG, IL18, HMGB1, TNF, NFKB1 |
| *histamine CTD* 00006100 | 1.03E-13 | 8.26E-11 | IL10, CXCL8, IFNG, IL18, TNF, NFKB1 |
| *terbutaline CTD* 00006840 | 3.80E-13 | 2.04E-10 | IL10, CXCL8, IFNG, TNF, NFKB1 |
| *PD98059 CTD* 00003206 | 2.15E-12 | 8.66E-10 | IL10, CXCL8, IFNG, IL18, TNF, NFKB1, IL17A |
| *aspirin CTD* 00005447 | 3.20E-12 | 1.03E-09 | IL10, CXCL8, IFNG, IL18, NLRP3, TNF, NFKB1, IL17A |
| 4 − *Methylhistamine CTD* 00001201 | 9.34E-12 | 2.48E-09 | IL10, IFNG, IL18, TNF |
| *TITANIUM DIOXIDE CTD* 00000489 | 1.08E-11 | 2.48E-09 | IL10, CXCL8, IFNG, TNF, NFKB1, IL17A |
| *Pyrrolidine dithiocarbamate CTD* 00001021 | 1.89E-11 | 3.74E-09 | CXCL8, IFNG, HMGB1, TNF, NFKB1, IL17A |
| *Caffeic acid phenethyl ester CTD* 00002352 | 2.34E-11 | 3.74E-09 | CXCL8, IFNG, IL18, TNF, NFKB1 |
| *sulfasalazine CTD* 00006719 | 2.34E-11 | 3.74E-09 | IL10, CXCL8, IFNG, TNF, NFKB1 |

**Table 8. Representing validated hub genes from RNA-seq datasets for PTB, datasets—GSE54992, GSE19442.**

| Gene Symbol | adj.P.Val | P.Value | logFC |
|---|---|---|---|
| CXCL8 | 4.530000e-03 | 4.520000e-04 | 1.808 |
| NLRP3 | 1.090000e-03 | 7.600000e-05 | 1.600 |
| TNF | 5.620000e-04 | 3.330000e-05 | 2.019 |
| IL10 | 1.070000e-04 | 4.190000e-06 | 2.904 |
| NFKB1 | 2.430000e-09 | 2.890000e-12 | 2.451 |

**Table 9. Representing validated hub genes from RNA-seq datasets for PD, datasets—GSE20295, GSE22491.**

| Gene Symbol | adj.P.Val | P.Value | logFC |
|---|---|---|---|
| IFNG | 0.0193 | 0.010738 | 0.705038 |
| IL18 | 0.000812 | 6.050000e-05 | 1.067785 |
| IL10 | 0.0193 | 0.011699 | 0.604016 |
| CXCL8 | 0.0193 | 0.009354 | 0.772296 |
| NFKB1 | 0.0193 | 0.011919 | 0.823916 |
| HMGB1 | 0.000003 | 1.810000e-08 | 0.916204 |

**Table 10. Representing validated hub genes from RNA-seq datasets for RA, datasets—GSE23561, GSE157047.**

| Gene Symbol | adj.P.Val | P.Value | logFC |
|---|---|---|---|
| IL10 | 0.00914 | 0.00182 | 0.829 |
| TNF | 1.150000e-10 | 7.180000e-11 | 1.625 |
| NFKB1 | 0.01220 | 0.00249 | 0.791 |
| IL18 | 3.230000e-17 | 1.370000e-17 | 1.780 |
| NLRP3 | 3.810000e-06 | 7.170000e-07 | 2.229 |
| CXCL8 | 1.480000e-03 | 4.980000e-04 | 2.148 |
| HMGB1 | 3.130000e-13 | 2.020000e-14 | 1.137 |

**Table 11. Representing validated hub genes from RNA-seq datasets for CKD, datasets—GSE66494, GSE15072, GSE141295.**

| Gene Symbol | adj.P.Val | P.Value | logFC |
|---|---|---|---|
| IL10 | 1.830000e-09 | 1.080000e-11 | 2.057 |
| NFKB1 | 3.110000e-07 | 5.280000e-09 | 0.745282 |
| TNF | 3.080000e-03 | 4.010000e-04 | 1.925763 |
| HMGB1 | 1.520000e-10 | 5.250000e-13 | 2.649 |
| CXCL8 | 1.060000e-06 | 2.600000e-08 | 4.136 |
| NLRP3 | 3.310000e-06 | 1.050000e-07 | 2.881 |

**Table 12. Representing validated hub genes from RNA-seq datasets for CVD, datasets—GSE51878, GSE141910.**

| Gene Symbol | adj.P.Val | P.Value | logFC |
|---|---|---|---|
| NFKB1 | 3.450000e-21 | 6.610000e-22 | 0.567 |
| IL10 | 0.000425 | 0.000094 | 1.296 |
| TNF | 7.020000e-18 | 1.730000e-13 | 1.201 |
| IFNG | 1.180000e-12 | 0.000094 | 2.438 |

**Table 13. Representing validated hub genes from RNA-seq datasets for LC, datasets—GSE42826, GSE30219.**

| Gene Symbol | adj.P.Val | P.Value | logFC |
|---|---|---|---|
| NLRP3 | 1.66e-04 | 4.83e-06 | 0.832 |
| HMGB1 | 1.28e-06 | 1.74e-07 | 0.729 |
| IL17A | 4.80e-06 | 4.37e-07 | 2.256 |
| IL18 | 2.89e-02 | 9.52e-03 | 0.716 |
| CXCL8 | 6.97e-06 | 6.64e-07 | 2.256 |
| TNF | 1.13e-03 | 3.20e-04 | 0.583 |

**Table 14. Representing validated hub genes from RNA-seq datasets for DM, datasets—GSE92724, GSE236746.**

| Gene Symbol | adj.P.Val | P.Value | logFC |
|---|---|---|---|
| IL18 | 1.100000e-09 | 2.300000e-12 | 4.155922 |
| TNF | 2.800000e-05 | 3.110000e-07 | 6.565250 |
| CXCL8 | 9.990000e-01 | 5.010000e-01 | 0.702678 |
| NLRP3 | 0.00751 | 0.000028 | 2.98 |

the degree value, five hub genes (CXCL8, NFKB1, TNF, IFNG, IL10) are selected from generic PPI. Regarding the PPI networks, the genes are linked among PTB and NCDs. On behalf of this study, tumor necrosis factor (TNF) and cytokine genes have been linked to PTB and NCDs [95]. The hub genes also revealed the presence of highly interconnected modules. Among the hub genes, IL-10 hub genes may help to discover targeted therapies for neurodegenerative disease because IL10 reduces TNF-$\alpha$ production in PD and increases Brain-derived neurotrophic factor (BDNF) levels [96].

On the other hand, in tissue-specific PPI, the NFKB1 gene interacts with the most proteins (subnetwork1), which regulates the infection response (from Table 2) in TB. In active TB patients, NFKB1 regulation is consistently up-regulated and regulates the transcription of genes related to both antiapoptotic responses and pro-inflammatory [97]. However, the prevalence of (auto)inflammatory issues among carriers indicates that modifying NFKB1 could intensify existing complications or induce fresh ones, thereby deteriorating symptoms and overall well-being. Besides, NFKB1 plays a crucial role in regulating immune responses. Targeting it could disrupt immune homeostasis, leading to further immune dysregulation and increased susceptibility to infections or autoimmune diseases [98].

Furthermore, the regulation of nine common genes is also confirmed by analyzing the gene regulatory network based on the performance of TF genes and miRNAs. After analyzing gene-miRNA interaction, the HMGB1 gene interconnects with the highest number of miRNA with a 171-degree value. And hsa-mir-34a-5P miRNA is interconnected with 4 genes (TNF, IL10, NFKB1, and HMGB1), which is identified as the most integrated miRNA with human diseases investigated by a study [99]. However, TCF7, NFIC, and TRIM22 are the most interacted TF in the TF-gene interaction network. TF genes serve as regulators for the gene expression that may cause the generation of cancer cells [100]. Besides the PPI network, TNF and NFKB1 genes interacted with the highest amount of TF with 20 and 17-degree values, respectively, in the TF-gene interaction network. A meta-analysis reveals that the -94ins/del polymorphism within the NFKB1 promoter is linked to cancer susceptibility, with potential ethnic-specific associations. This highlights the role of NF-$\kappa$B signaling in oncogenesis and proposes it as a promising therapeutic target [101]. TCF7 TF contributes to pulmonary infection and assists in

tissue regeneration and repair after severe lung damage [102]. On the other hand, NFIC TF is related to cancer if its development is disrupted [103].

GO terms and pathways were identified based on P-values, considering significance when the P-value was <0.05. Results were deemed statistically significant if the P-value was below 0.05 [104]. From the result of GO pathways, Cellular response to lipopolysaccharide, inflammatory response, positive regulation of cytokine production, cytokine-mediated signaling pathway, etc., are the top GO terms in biological processes (Table 6). Cytokine activity, cytokine receptor binding, receptor-ligand activity, etc., are found in molecular function (Table 6). Lastly, cellular components include the cytoplasmic vesicle lumen, secretory granule lumen, etc. GO (Table 6). On the other hand, Inflammatory bowel disease, yersinia infection, malaria, influenza A, pertussis, rheumatoid arthritis, IL-17 signaling pathway, amoebiasis, chagas disease, cytokine-cytokine receptor interaction were the top 10 pathways extracted from the KEGG database to analysis pathways (Table 7). IL-10 Anti-inflammatory Signaling Pathway Homo sapiens h il10Pathway, NF-$\kappa$B Signaling Pathway Homo sapiens h nfkbPathway, NF$\kappa$B activation by Nontypeable Hemophilus influenzae Homo sapiens h nthiPathway are highly connected with genes in Bio-Carta (Table 7). From the wikiPathwas pathway, IL-18 signaling pathway WP4754 is the highest-connected gene pathway (Table 7). Lastly, from the Reactome pathway, signaling By Interleukins R-HSA-449147, cytokine Signaling In Immune System R-HSA-1280215, and immune System R-HSA-168256 were found with highly connected genes (Table 7). The development of PTB is intricately linked to inflammatory responses, providing a potential avenue for therapeutic intervention in critically ill PTB patients in the intensive care unit (ICU). Elevated levels of interleukins (IL-1, IL-4, IL-6, IL-10, IL-12) in ICU PTB patients underscore the significant role of these cytokines in active TB disease. Genetic associations with cytokines (IFNG, TNF, IL17A, IL10) impact cellular immunity, disease progression, and treatment outcomes. Additionally, IL-1, IL-10, IL-17, and IL-18, as key cytokine family members, play crucial roles in regulating immunological and inflammatory responses. Understanding these complex interactions suggests a potential for targeted interventions to modulate inflammation, thereby improving outcomes in ICU PTB patients. This comprehensive insight into the interplay between cytokines and TB pathogenesis emphasizes the importance of considering inflammatory modulation as a therapeutic strategy in managing critically ill PTB cases [105–108].

The DSigDB database was utilized to identify the drug component. The top 10 drug components- bay 11–7082 CTD 00003959, histamine CTD 00006100, terbutaline CTD 00006840, PD 98059 CTD 00003206, aspirin CTD 00005447, 4-Methylhistamine CTD 00001201, TITANIUM DIOXIDE CTD 00000489, Pyrrolidine dithiocarbamate CTD 00001021, Caffeic acid phenethyl ester CTD 00002352, and sulfasalazine CTD 00006719 were identified based on P-values and adjusted P-values. Aspirin (CTD 00005447), a well-known ERK inhibitor, inhibits inflammation, protects humans from neurodegenerative diseases, and is a suggested drug for RA [109, 110]. Among the drug components, Aspirin can produce lipoxins that have beneficial pro-resolving effects, effectively controlling disease progression in TB by mitigating hyper-inflammatory responses [111]. Besides, Histamine is crucial in the body's immune response against infections [112]. BAY 11–7082 hinders the movement of NFKB into the nucleus cell by inhibiting I-$\kappa$B kinase-$\beta$ and suppressing the activation of the NLRP3 inflammasome [113]. Sulfasalazine is commonly used to treat rheumatoid arthritis and reduces TNF and prostaglandin synthesis, providing therapeutic benefits [114]. Titanium dioxide-based drugs can reduce the risk of tumorigenesis and enhance cancer treatment [115]. The effect of a drug on the body and how well it works depend on how well it binds to the specific proteins and how much it alters the network of protein-protein and protein-chemical interactions. The concentration, strength, and distribution of target proteins in tissues affect the effectiveness of the medicine [116]. Chemokines and cytokines are crucial for organizing and guiding the immune cells into

the lungs during M. TB infection. Proper recruitment and positioning of these cells are essential for controlling bacterial growth without causing harmful inflammation. Examination of cytokine/chemokine interaction and its conditional significance in infection and disease progression to understand TB pathogenesis [117].

In summary, the research provided a comprehensive analysis of the genetic underpinnings of various diseases, emphasizing the interconnected nature of genetic, proteomic, and environmental factors in disease pathology. This holistic approach opens new avenues for understanding disease mechanisms and developing targeted therapies. But, this study exclusively focused on bioinformatics analyses. Besides, we collect data from one source to identify drug components that may be influencing performance.

## Conclusion

This bio-informatics study focused on the links, pathways, and medication components between PTB and NCDs, such as lung cancer, diabetes mellitus, Parkinson's disease, silicosis, chronic kidney disease, cardiovascular disease, and rheumatoid arthritis. It is discovered that nine genes (NFKB1, IFNG, TNF, HMGB1, CXCL8, NLRP3, IL18, IL17A, and IL10) are connected to the development and spread of the NCDs mentioned above when a person is already diagnosed with PTB. Gene ontology and pathway analysis were utilized to discover the biological processes and pathways that these genes were engaged in. The fact that these genes were discovered in several pathways related to inflammatory response, immunological response, and cytokine signaling implies that they play a role in how various disorders begin.

Additionally, five hub genes (NFKB1, TNF, CXCL8, NLRP3, and IL10) were identified as potential therapeutic targets after analysis of a network of protein-protein interactions. These genes were selected as they have many connections and are essential in the network. The findings of this study provide critical information regarding potential therapeutic targets that might be employed to treat common ailments. This research has significance for bio-informatics research, clinicians, drug discovery, and other complicated fields of study. These potential therapeutic targets may undergo several investigations, such as in vitro and in vivo. The development of novel drugs that can target the mentioned genes in this study to treat the discussed common disorders may benefit long-term chronic patients.

## Supporting information

**S1 File. Data availability statement.** This document includes the data sources and accession numbers for each dataset.
(DOCX)

## Author Contributions

**Data curation:** Amira Mahjabeen.

**Formal analysis:** Md. Zahid Hasan, Md. Tanvir Rahman.

**Investigation:** Amira Mahjabeen.

**Methodology:** Amira Mahjabeen, Md. Tanvir Rahman, Md. Aminul Islam.

**Project administration:** Md. Zahid Hasan, Md. Tanvir Rahman, M. Shamim Kaiser.

**Resources:** Md. Aminul Islam.

**Software:** Amira Mahjabeen.

**Supervision:** Md. Zahid Hasan, Md. Tanvir Rahman, M. Shamim Kaiser.

**Validation:** M. Shamim Kaiser.

**Visualization:** Md. Aminul Islam.

**Writing – original draft:** Md. Tanvir Rahman, Risala Tasin Khan.

**Writing – review & editing:** Md. Tanvir Rahman, Risala Tasin Khan.

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
