## [Decision Letter · Decision Letter 0]

23 Apr 2024

PONE-D-23-41835Genetic Insights into the Connection Between Pulmonary TB and Non-Communicable Diseases: An Integrated Analysis of Shared Genes and Potential Treatment TargetsPLOS ONE

Dear *Dr. *Rahman,

Thank you for submitting your manuscript to PLOS ONE. After careful consideration, we feel that it has merit but does not fully meet PLOS ONE’s publication criteria as it currently stands. Therefore, we invite you to submit a revised version of the manuscript that addresses the points raised during the review process.

We look forward to receiving your revised manuscript.

Kind regards,

Ravikanth Nanduri, Ph. D.

Academic Editor

PLOS ONE

2. We note that Figure 1 in your submission contain copyrighted images. All PLOS content is published under the Creative Commons Attribution License (CC BY 4.0), which means that the manuscript, images, and Supporting Information files will be freely available online, and any third party is permitted to access, download, copy, distribute, and use these materials in any way, even commercially, with proper attribution. For more information, see our copyright guidelines: http://journals.plos.org/plosone/s/licenses-and-copyright.

3. Please remove your figures from within your manuscript file, leaving only the individual TIFF/EPS image files, uploaded separately. These will be automatically included in the reviewers’ PDF.

Reviewers' comments:

Reviewer's Responses to Questions

**Comments to the Author**

1. Is the manuscript technically sound, and do the data support the conclusions?

Reviewer #1: Partly

Reviewer #2: Partly

2. Has the statistical analysis been performed appropriately and rigorously? 

Reviewer #1: Yes

Reviewer #2: Yes

3. Have the authors made all data underlying the findings in their manuscript fully available?

Reviewer #1: Yes

Reviewer #2: Yes

4. Is the manuscript presented in an intelligible fashion and written in standard English?

Reviewer #1: No

Reviewer #2: Yes

5. Review Comments to the Author

Reviewer #1: I wanted to comment upon the language of the paper, I can fully understand it is difficult for a non-native speaker to be fluent in English, but I expect the authors to improve the language of the manuscript thoroughly as the grammatical errors dilute the actual potential of the manuscript.

Reviewer #2: Authors have put in a commendable effort using bioinformatics in addressing one of the challenging questions in treatment of patients with comorbidities. In this article authors have investigated the genetic association between PTB and 7 NCDs. They show that genes such as TNF, IL10, NLRP3, IL18, IFNG, HMGB1, CXCL8, IL17A, and NFKB1 discovered were shared between TB and the NCDs, and five hub genes (NFKB1, TNF, CXCL8, NLRP3, and IL10) were found to be common among them. Moreover, they also suggested few drug candidates for the treatment. Although, further investigation is suggested for better treatment outcomes.

The manuscript is very well written.

However, I have one major concern:

Authors have shown NFKB1 as a potential target. Given the pleiotropic role of NFKB1, approach to target NFKB1 rather becomes interesting and the question of specificity immediately arises. It is vital to consider the underlying mechanisms of disease progression to obtain a required beneficial outcome while targeting NFKB1 in patients with comorbidities. Consider following situations involved in comorbidities:

a. In autoimmune diseases like RA immune system needs to be suppressed to alleviate the disease while in PTB need of effective immune system is essential.

b. In cancer, while rouge cells are required to be controlled, immune system needs bolstering.

Also, a more nuanced approach is advised such as targeting of subsets of NF-κB activities which, hopefully, will target pathogenic processes while leaving physiological processes largely intact.

Could the authors discuss the complication of targeting NFKB1, while it is one of the major hub genes as per their study is concerned.

Minor correction

Line no. 378- On the other hand, In tissue-specific PPI, the NFKB1 gene interacts with. Small case, ‘in’.

6. PLOS authors have the option to publish the peer review history of their article (what does this mean?). If published, this will include your full peer review and any attached files.

Reviewer #1: **Yes: **SAKSHI SINGH

Reviewer #2: No

---

## [Author Response · Author response to Decision Letter 0]

2 Aug 2024

Response to Reviewers file is attached

---

## [Decision Letter · Decision Letter 1]

30 Sep 2024

Genetic Insights into the Connection Between Pulmonary TB and Non-Communicable Diseases: An Integrated Analysis of Shared Genes and Potential Treatment Targets

PONE-D-23-41835R1

Dear Dr. Rahman,

We’re pleased to inform you that your manuscript has been judged scientifically suitable for publication and will be formally accepted for publication once it meets all outstanding technical requirements.

Kind regards,

Jinhui Liu

Academic Editor

PLOS ONE

Additional Editor Comments (optional):

I think this manuscript was well organized and it could be accepted.

Reviewers' comments:

Reviewer's Responses to Questions

**Comments to the Author**

1. If the authors have adequately addressed your comments raised in a previous round of review and you feel that this manuscript is now acceptable for publication, you may indicate that here to bypass the “Comments to the Author” section, enter your conflict of interest statement in the “Confidential to Editor” section, and submit your "Accept" recommendation.

Reviewer #1: All comments have been addressed

Reviewer #2: All comments have been addressed

2. Is the manuscript technically sound, and do the data support the conclusions?

Reviewer #1: Yes

Reviewer #2: Yes

3. Has the statistical analysis been performed appropriately and rigorously? 

Reviewer #1: Yes

Reviewer #2: Yes

4. Have the authors made all data underlying the findings in their manuscript fully available?

Reviewer #1: Yes

Reviewer #2: Yes

5. Is the manuscript presented in an intelligible fashion and written in standard English?

Reviewer #1: Yes

Reviewer #2: Yes

6. Review Comments to the Author

Reviewer #1: (No Response)

Reviewer #2: (No Response)

7. PLOS authors have the option to publish the peer review history of their article (what does this mean?). If published, this will include your full peer review and any attached files.

Reviewer #1: No

Reviewer #2: No

---

## [Editor Report · Acceptance letter]

9 Oct 2024

PONE-D-23-41835R1 

PLOS ONE

Dear Dr. Rahman, 

I'm pleased to inform you that your manuscript has been deemed suitable for publication in PLOS ONE. Congratulations! Your manuscript is now being handed over to our production team.

Kind regards, 

on behalf of

Dr. Jinhui Liu 

Academic Editor

PLOS ONE